# Epidemiology of Suicide Mortality in Paraguay from 2005 to 2019: A Descriptive Study

**DOI:** 10.3390/ijerph21030277

**Published:** 2024-02-28

**Authors:** Ethel Santacruz, Derlis Duarte-Zoilan, Gilda Benitez Rolandi, Felicia Cañete, Dins Smits, Noël C. Barengo, Guillermo Sequera

**Affiliations:** 1Department of Noncommunicable Disease Surveillance, Ministry of Health and Social Welfare, Asunción 001221, Paraguay; erosantacruz@gmail.com (E.S.); derliszoilan@gmail.com (D.D.-Z.); gbenitezrolandi2@gmail.com (G.B.R.); feliciacanete@gmail.com (F.C.); 2Department of Public Health, Faculty of Medicine, National University of Asunción, Asunción 011221, Paraguay; 3Faculty of Medicine, Riga Stradins University, LV-1007 Riga, Latvia; dins.smits@rsu.lv; 4Department of Medical Education, Herbert Wertheim College of Medicine, Florida International University, Miami, FL 33199, USA; nbarengo@fiu.edu; 5School of Medicine, National University of Mar del Plata, Mar del Plata 7600, Argentina

**Keywords:** suicide, trends, mental health, Paraguay, potential years of life lost

## Abstract

Suicide is an important public health problem, fundamentally affecting the younger population and responding to multiple biological, psychological, and social causes. The objective of this study was to characterize changes in suicide mortality, suicide methods, and years of potential life lost from 2005 to 2019 in Paraguay. This observational, descriptive study used data from the Vital Statistics Information Subsystem of the Ministry of Public Health and Social Welfare. The average mortality rate from suicide was 4.9 per 100,000 inhabitants, with an increase from 4.2 between 2005 and 2009 to 5.8 from 2014 to 2019. Suicide was more common in men (75%) than in women. In men, the highest mortality rate was observed among those 20–24 years old, whereas in women, the ages most affected were the 15–19-year-old age group. The most-used method for suicide was hanging. The most frequent place of suicide occurrence was at home (73%). The seasonality of suicide occurrence showed a slight increase in the spring–summer months compared with autumn–winter (53% vs. 47%). The rate of potential years of life lost statistically significantly increased from 2005 to 2019. Public health measures need to be implemented to investigate the underlying reasons and implement interventions in the population to decrease suicide mortality in Paraguay.

## 1. Introduction

At a global level, suicide is a complex public health problem, fundamentally affecting the younger population and responding to multiple biological, psychological, and social causes [1,2]. It presents a significant global burden of disease for all societies [3,4]. It has been estimated that suicides were responsible for over 700,000 deaths worldwide in 2019 [5]. This represents approximately 1.3% of all deaths globally, making it the 17th leading cause of death in 2019 [5]. Furthermore, in 2019, the global age-standardized suicide mortality rate was estimated to be 9.0/100,000 per population, with a higher rate in men than women (12.6 in males vs. 5.4 in females) [4]. Even though a few countries have observed a decreasing trend in suicide mortality, data from most parts of the world show an increasing trend in deaths due to suicide [4]. Specifically, North, Central, and South America have experienced a significant increase in suicide mortality over the past 20 years [4]. Ilic et al. reported that, between 2000 and 2019, suicide mortality increased from 3.6/100,000 to 6.2/100,000 in Paraguay.

In Paraguay, although cardiovascular diseases and cancer continue to be the leading causes of death in adults, external causes such as traffic accidents and suicide are becoming more frequent every day and represent a significant burden of premature mortality [6,7]. In recent years, there has been a linear increase in deaths by suicide, especially among young people in Paraguay, but the characteristics of these events are still unknown [8].

The methods used for suicide vary according to country, culture, time, and demographic characteristics, such as biogeographical origin, sex, age, and rural or urban environment [9]. The most common suicide method used in Paraguay was intentional self-inflicted injury by hanging, strangulation, or suffocation (all 67.6%) [8]. Suicide also presents large geographical and temporal variations according to country [10,11,12]. Previous studies have suggested a relationship between suicide rates and other factors, such as economic, environmental, and temporal factors such as seasonality [10,12,13]. Unfortunately, epidemiological data on years of potential life lost due to suicide, changes in suicide methods, and seasonal patterns over time, as well as trends of suicide mortality in different states of Paraguay, are scant.

Regrettably, there is a paucity of epidemiological data regarding the years of potential life lost owing to suicide, shifts in suicide methods across time periods, seasonal fluctuations, and trends in suicide mortality rates between the various states that comprise Paraguay. While understanding the dynamics behind suicide completion at a granular level could help to direct preventative efforts, limitations currently exist as to the availability and breadth of relevant statistics. It is hoped that public health authorities may seek to bolster record-keeping and analytical efforts going forward, to build a more comprehensive picture of how this tragic act impacts the population in terms of factors like demographics, geography, and temporality. With a more robust evidence base, stakeholders could potentially devise time- and location-sensitive countermeasures to reduce suicide rates in a targeted manner.

The objective of this study was to describe changes suicide mortality, suicide methods, and years of potential life lost from 2005 to 2019 in Paraguay. An understanding of changes over time is important to guide public health professionals in developing, implementing, and monitoring public policies to decrease suicide attempts and suicide mortality.

## 2. Material and Methods

### 2.1. Study Design and Data

A descriptive, observational study was conducted using validated records from the General Directorate of Strategic Health Information of the Paraguayan Ministry of Public Health and Social Welfare. This institution is officially responsible for collating national health and disease statistics within Paraguay. The records accessed spanned the period from 2005 to 2019. Data were sourced directly from the validated records held by the General Directorate, being the government body statutorily tasked with the collection and maintenance of healthcare and morbidity data at a nationwide level in Paraguay. To define all cases of death by suicide occurring in Paraguay across the study period, the International Classification of Diseases 10th Revision (ICD-10) codes X60 through X84 were referenced. These numerical codes within the ICD-10 classification system specifically relate to and encompass intentional self-harm, also known as suicide. By limiting the analysis to only those records coded as such, the researchers were able to accurately identify and extract all instances of completed suicide recorded in Paraguay between 2005 and 2019 inclusive. All data between 2005 and June 2019 were analyzed, as at the time of the analysis, this was the time span where mortality data were available, cleaned, validated, and fully entered into the registry.

### 2.2. Measures of Suicide Mortality and Characteristics

The data collected from death certificates and utilized in this research included several key variables. Demographic information such as the age and sex of the deceased was recorded, with sex categorized as either male or female. Additional details pertaining to the date of death and level of attained education provided important context. The primary cause of mortality was also documented. The state of residence of the individual prior to their passing was noted, as was the location where the suicide took place. Whether this was within the place of residence or elsewhere was distinguished. The geographic area where the person resided was specified. Finally, the specific means by which the victim ended their life, or their suicide method, was identified. This suite of variables from official documentation served to facilitate an insightful analysis of trends within the departed population. Age was categorized into the following age groups: (i) <10 years, (ii) 10–14 years-old, (iii) 15–24 years-old, (iv), 25–44 years-old, (v) 45–64 years-old, and (vi) ≥65 years-old. Educational level was divided into none, primary, high school, college/university, and unknown. Suicide method was defined according to ICD-10 codes and were as follows: poisoning (X60–X69), intentional self-inflicted injury by hanging, strangulation, or suffocation (X70), drowning (X71), firearm firing (X72–X74), and other (X76–X84). Information on the event site was defined as at home, at healthcare institutions, public location/area, and others. The last recorded residence of the suicide victim was dichotomized into rural versus urban. The season of the suicide occurrence was recorded as spring/summer and autumn/winter. Finally, the time between 2005 and 2019 was divided into three intervals (i) 2005–2009, (ii) 2010–2014, and (iii) 2015–2019.

### 2.3. Data Analysis

The data were analyzed using Microsoft Excel^®^ and Epi-Info7 version 7.2.1.1. The data were analyzed according to time interval, age-groups, sex, and state. For the analysis of general trends in suicides, specific rates calculated by the number of deaths in each age group and sex at the national and departmental levels were used, divided by the annual population of the corresponding five-year period for each age group. Direct standardization was used to calculate age-specific mortality rates for the different time intervals. Population estimates were obtained from the Paraguayan General Office of Statistics, Survey and Censuses. National projection of population by sex and disaggregated by age group (2000–2050) based on the 2002 Census was used for the time interval of 2006–2014 [14,15]. Population estimates starting from 2015 used data from the 2012 population census and projections for 2000–2025 [16]. The world population was used as the WHO standard population for rate adjustments (2000–2025). The years potential of life lost (YPLL) were calculated considering 0 years as a lower limit for the cut-off point and 70 years as the upper age limit for both sexes per thousand inhabitants. Suicide mortality rates were calculated per hundred thousand inhabitants as a denominator. The trend of suicide mortality at the national level from 2005 to 2019 was analyzed with the Joinpoint Trend Analysis Software (version 4.8.0.1) [17]. The software took suicide trend data at the national level (increases or decreases) and the rate or rate of annual average change and fit the simplest joinpoint model that the data allowed. The minimum and maximum number of joinpoints were supplied by the investigators. The program started with the minimum number of joinpoints and tests, later determining whether more joinpoints were statistically significant and had to be added to the model up to that maximum number. This enabled us to test whether an apparent change in trend was statistically significant. The tests of significance used the Monte Carlo Permutation method.

## 3. Results

### 3.1. Sociodemographic Characteristics of Deaths Due to Suicide

Table 1 shows the sociodemographic characteristics of deaths due to suicide in Paraguay between 2005 and 2019. In total, 4776 deaths by suicide were recorded, corresponding to 1.4% of all deaths recorded. A 62.1% increase between the first-time interval (2005–2009) and the last one (2015–2019) was observed. The percentage of men among all suicides increased from 71.2% (*n* = 884) (2005–2009) to 75% (*n* = 1517) in 2015–2019. No statistically significant changes were observed in the distribution of age groups according to the three intervals. Most suicides were recorded in the 15- to 24-year-old population, followed by the 25- to 44-year-old age group. There was an increase in the percentage of cases with high school or university degrees between 2005 and 2019. While 13.1% (*n* = 163) of all suicide cases between 2005 and 2010 had high school and 3.9% (*n* = 48) had a university degree, the corresponding percentages of suicide cases with a high school degree (16.6%; *n* = 336) and university degree (5.3%; *n* = 108) was higher in 2015–2019. There seemed to be a change in the site where suicide occurred from 2005 to 2019. The percentage of suicides that occurred in healthcare services decreased from 23% (*n* = 283) (2005–2009) to 14% (*n* = 207) (2010–2014) and 10% (*n* = 199) in 2015–2019. Suicides at home increased from 63% (*n* = 779) (2005–2009) to 73% (*n* = 1471) (2015–2019). While during 2005–2009, every second recorded suicide was by hanging (50%; *n* = 622), hanging accounted for three out of four suicides in 2015–2019. During the same time interval, the share of suicide by poisoning decreased from 19% to 3.5% (*n* = 71) and the use of firearms from 24% (*n* = 298) to 15% (*n* = 303). An increase in the prevalence of suicide in the urban region (from 72% to 77%) was observed between 2005 and 2019. Finally, 56–58% of suicides occurred the during spring/summer period during 2005–2019.

### 3.2. Standardized Mortality Rate for Suicides

Of the 18 health regions, 12 experienced increases in suicide rates during the 2005–2019 period (Figure 1). The regions that had increases in suicide rates above the national average were Presidente Hayes, Central, Boquerón, Caazapá, Capital, Amambay, Canindeyú, and Concepción.

The average suicide mortality rate nationwide between 2005 and 2019 was 4.9/100,000, with a gradual increase per five-year period, from 4.3/100,000 (2005–2009) to 5.7/100,000 between 2015 and 2019. Table 2 presents the standardized mortality rate for suicides according to 5-year time intervals in Paraguay between 2005 and 2019. The age-standardized mortality rate for suicides increased in men from 6.1/100,000 (2005–2009) to 8.5/100,000 during 2015–2019. The corresponding increase in women was much lower (from 2.4/100,000 in 2005–2009 to 2.9/100,000 in 2015–2019). Also, the gap in mortality rates for suicide increased between men and women between 2005 and 2019.

A joinpoint analysis revealed that suicide rates linearly increased by 3.82% per year in Paraguay between 2005 and 2019 (Figure 2; *p*-value < 0.001).

### 3.3. Rate of Years of Potential Life Lost by Suicide

During the 15 years of observation, 172,612 years of potential life were lost due to suicide (Table 3). The YPLL rate due to suicide increased in the Paraguayan population during the study period. While the YPLL was 1.5/1000, the corresponding rates for 2010–2014 and 2015–2019 were 1.8/1000 and 2.1/1000. While the YPLL rate increased in men from 2.1/1000 (2005–2009) to 3.02/1000 during 2015–2019, it remained unchanged in women (1–1.1/1000) during the three time intervals.

## 4. Discussion

Our data revealed that there was a linear increase in suicide mortality in Paraguay between 2005 and 2019. The age-standardized mortality rate for suicides increased more in men than in women, widening the gap in suicide mortality rates between men and women between 2005 and 2019. The YPLL due to suicide increased in men by 50%, but remained stable in women. Three out of four suicides occurred by hanging, whereas suicide by poisoning and firearms showed a decreasing trend.

Even though the suicide mortality rate has increased, it is still lower than those reported from Argentina, Brazil, Chile, and Uruguay [4]. However, a recent study that used preliminary data predicted a further increase in suicide mortality in Paraguay and that that the expected number of suicides in the following years from 2023 to 2027 will range between 462 and 530 per year [8]. There are several reasons that may explain the increase in suicide mortality rate in Paraguay. First, this increase in suicide mortality may be due to an increase in mood and alcohol use disorders. These disorders are the most important modifiable predictors of suicide and [18,19] both have shown increases in the region [20,21,22]. Moreover, recent scientific evidence has suggested that acute alcohol use increases the risk of suicide attempts and that the risk of suicidal ideation, suicidal attempts, and death by suicide are each 2–3 times higher in people with alcohol use disorders compared with the general population [23,24]. A small part of the higher mortality rates observed may be explained by improved methods to differentiate suicide mortality from injury- and accident-related mortality, as some suicide deaths tend to be reported as accidental or underdetermined due to shame, stigma, or lack of evidence [25]. To some extent, we cannot rule out the cohort effect. The variation over time among the different age cohorts of the suicide cases may mask or distort effects which might be present in smaller, more homogeneous, constituent sub-cohorts, as they may experience different rates of outcomes.

The data indicating that males had a higher suicide mortality rate than women are consistent with the scientific literature [26,27]. The gender differences in suicide intention or mortality rates are known as the “Gender Paradox” [27]. A systematic review and meta-analysis of longitudinal studies revealed that women had a two-fold higher risk of suicide attempts than men, whereas the mortality rate of suicide was 2.5 times higher in men compared with women [28,29]. Another study revealed that women had higher rates of suicidal thinking, non-fatal suicidal behavior, and suicide attempts compared with men [30]. Thus, it could be said that women seem to exhibit less serious intent to die than men [31]. Probably, the most important reason for the observed differences between suicide attempts and suicide mortality between men and women is the method of suicide used. According to scientific evidence, men tend to choose more lethal suicide methods (firearms, hanging, and asphyxiation for instance), while women were more prone to overdose on medications or drugs than men [32]. Our data regarding higher suicide mortality rates in those with lower levels of education are consistent with previous studies. Several studies in different populations have revealed an inverse association between educational levels and suicide mortality rates in different populations [33,34,35,36,37]. It has been suggested that this may be due to better opportunities for being employed for people with higher levels of education, resulting in a higher level of social integration and lower suicide rates [33,37,38], while having lower levels of education may increase the chance of unemployment, leading to less responsibilities, more narrow networks, and less social ties [37].

Our data revealed a marked gap in suicide deaths among urban and rural populations, which may be related to several factors, ranging from political to anthropological factors. Several epidemiological studies have described these urban–rural variations. However, data on the associations between urban–rural areas and suicide have been contradictory [39,40,41,42,43,44]. While some studies have reported statistically significantly higher suicide rates in urban compared with rural areas, some have revealed the opposite, and others have not found a difference in suicide between urban and rural areas. It has been proposed that stigmas surrounding mental health issues, social isolation, limited social support, socioeconomic conditions, lifestyles, and environmental factors can all play a role in influencing the wellbeing of rural communities [45]. Finally, a recent literature review on the inequalities of suicide mortality across urban and rural areas did not reveal an urban–rural pattern regarding urban–rural changes over time [46]. Future research may explore the associations between urban–rural areas and suicide mortality and its underlying determinants.

Even though suicide is a serious public health concern in Paraguay, the existing strategies remain limited. Efforts to mitigate the incidence of suicide predominately revolve around restricting access to the means of self-harm for individuals at risk, fostering social and emotional life skills in adolescents, ensuring the availability of integrated mental health services at the primary care level, and dismantling the stigma surrounding mental health—an obstacle often impeding individuals from seeking help. Recently, the Pan American Health Organization introduced a comprehensive strategy for enhancing mental health and preventing suicide across the Americas, focusing on six strategic lines of actions [47]: (i) fostering mental health leadership, governance, and multisectoral partnerships, and integrating mental health in all policies; (ii) enhancing the availability, accessibility, and quality of community-based services for mental health conditions, and supporting the advance of deinstitutionalization; (iii) promoting mental health promotion and prevention strategies and activities throughout the life course; (iv) integrating mental health and psychosocial support in emergency contexts; (v) strengthening data, evidence, and research; and (vi) making suicide prevention a national whole-of-government priority and building multisectoral capacity to respond to people affected by suicidal behaviors.

Another proposed community intervention to address suicide in the region is the Engaged Community Action for Preventing Suicide (ECAPS) Model, specifically designed for low-resource settings to confront systemic, societal, and individual level factors [48]. This model proves to be both feasible and effective in guiding the development of culturally relevant, community-level intervention. ECAPS adopts a community-based participatory approach, fostering cross-cultural partnerships, resource sharing, and shared decision-making power [49]. The seven key steps involve: (1) identifying the relevant determinants specific to the community associated with suicide risk; (2) assessing the resources available (and not available) in the community; (3) prioritizing determinants according to available resources; (4) developing a change strategy; (5) implementing the strategy; (6) evaluating outcomes; and (7) identifying next steps. The results of the ¡PEDIR! Program revealed that the ECAPS model is a feasible and effective framework for use in low-resource settings to guide the development of a culturally relevant community-level intervention to address the systemic, societal, and individual level factors that serve as barriers to suicide prevention [49].

We also would like to point out that little information exists on the prevalence of depression in the Paraguayan population. A study among a representative sample of medical students at the National University in Asuncion revealed that 19% suffered from depression [50]. Torales et al. analyzed, in 2016, all psychiatric consultations carried out over one year at the Psychiatry Unit of the Emergency Department at the National University of Asunción’s Clinical Hospital in Paraguay to describe the clinical and demographic characteristics of patients, and reported that approximately 17% of the patient consultations were related to depressive symptoms [51]. Finally, the Paraguayan burden of disease study found that mental behavioral disorders were the number one contributor to years of healthy life lost, with 46.4 years per 1000 inhabitants in 2019 [52]. There is a clear necessity to develop population studies to collect information on mental health disorders in the population to receive a more up-to-date information on the socio-demographic characteristics of mental health problems in Paraguay.

Naturally, our study has some limitations. Under-reporting of suicide may introduce bias in the assessment of suicide mortality. In addition, the quality of mortality statistics (coverage, accuracy, and completeness of data) may have introduced some bias in estimating suicide mortality. For instance, data on suicide mortality in the indigenous population were not available. The department of vital statistics reported that there is an underreporting of deaths that accounts up to 18% in the last available year (2019). Also, the validity of death certification for suicide is a major problem, as some suicides may be classified as undetermined intent, accidental, or violent deaths. The vital statistics department revealed that approximately 7% of deaths were incorrectly classified [53]. Moreover, one limitation of this study is that we did not have information on the medical conditions of the suicide cases, as the data were obtained from the Vital Statistics Information Subsystem, which only registers the basic cause of deaths.

Finally, information on cases of complex suicides such as two different suicide modes was not available.

## 5. Conclusions

In conclusion, the increase in suicide mortality in Paraguay is of concern and an important cause of the burden of premature death. Future analytical, observational studies should be conducted to identify what the most important predictors of suicide are. Also, more data are needed on the associations between social determinants of health and increasing suicide mortality trends. Moreover, a surveillance system monitoring the characteristics and trends of suicide in Paraguay needs to be implemented to assist political decision makers in developing policies and population-wide interventions to identify people at risk early in order to offer counseling and support. Finally, to decrease the observed suicide mortality, interdisciplinary collaboration is needed including health professionals, political decision makers, educators, and others.

## Figures and Tables

**Figure 1 ijerph-21-00277-f001:**
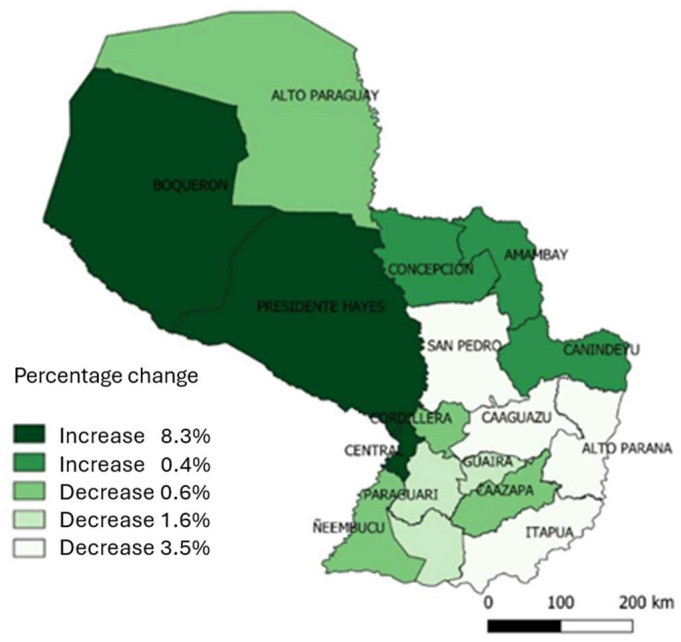
Crude suicide mortality rate per 100,000 inhabitants according to health regions and five-year period in Paraguay during 2005 and 2019.

**Figure 2 ijerph-21-00277-f002:**
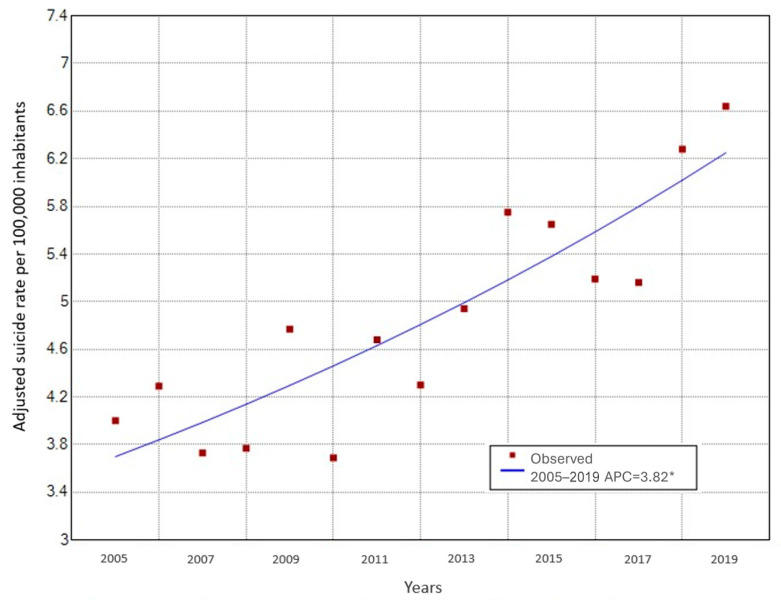
Average annual percentage change (APC) trend of suicide mortality 2005–2019. ***** Indicates that Annual Percent of Change (APC) is significantly different from zero at the alpha = 0.05 level. This figure is select model with 0 joinpoints.

**Table 1 ijerph-21-00277-t001:** Sociodemographic characteristics of deaths due to suicide in Paraguay between 2005 and 2019.

Five-Year Periods	2005–2009			2010–1014			2015–2019		
	No.	%	95%CI	No.	%	95%CI	No.	%	95%CI
Total	1241			1513			2022		
Sex									
Man	884	71.2	(68.6–73.6)	1096	72.4	(70.1–74.6)	1517	75.0	(73.0–76.8)
Woman	357	28.8	(26.3–31.3)	417	27.6	(25.3–29.8)	505	25.0	(23.1–26.9)
Age Group									
<10 years	24	1.9	(1.3–2.8)	10	0.7	(0.3–1.2)	2	0.1	(0.0–0.3)
10–14 years	63	5.1	(3.9–6.4)	74	4.9	(3.9–6.1)	83	4.1	(3.3–5.0)
15–24 years	447	36.0	(33.4–38.7)	567	37.8	(35.3–40.2)	714	35.5	(33.4–37.6)
25–44 years	406	32.7	(30.1–35.3)	508	33.8	(31.4–36.2)	661	32.9	(30.8–34.9)
45–64 years	202	16.3	(14.3–18.4)	236	15.7	(13.9–17.6)	380	18.9	(17.2–20.6)
>65 years old	99	8.0	(6.6–9.6)	107	7.1	(6.9–8.5)	172	8.6	(7.4–9.8)
Level of Education									
None	46	3.7	(2.7–4.9)	68	4.5	(3.5–5.6)	94	4.7	(3.8–5.1)
Primary	615	49.6	(46.7–52.3)	682	45.1	(42.5–47.5)	886	43.8	(41.6–45.9)
High school	163	13.1	(11.3–15.1)	217	14.3	(12.6–16.2)	336	16.6	(15.0–18.3)
University	48	3.9	(2.9–5.0)	79	5.2	(4.2–6.4)	108	5.3	(4.4–6.4)
Doesn’t know	369	29.7	(27.2–32.3)	467	30.9	(28.5–33.2)	598	29.6	(27.6–31.6)
Types of Methods									
Poisoning	230	18.5	(16.4–20.7)	95	6.3	(5.1–7.6)	71	3.5	(2.7–4.4)
Hanging	622	50.1	(47.3–52.9)	1021	67.8	(65.0–69.8)	1508	74.6	(72.6–76.4)
Drowning	30	2.4	(1.7–3.4)	29	1.9	(1.3–2.7)	42	2.1	(1.5–2.8)
Firearm	298	24.0	21.7–26.4	301	19.9	17.9–21.9	303	15.0	(13.5–16.6)
Other causes ^1^	61	4.9	(3.8–6.2)	67	4.4	(3.5–5.5)	98	4.9	(3.9–5.8)
Event site									
Health Services	283	22.8	(20.5–25.2)	207	13.7	(12.0–15.5)	199	9.8	(8.6–11.2)
Home	779	62.8	(60.0–65.4)	1069	70.7	(68.3–72.8)	1471	72.8	(70.7–74.6)
Public area	125	10.1	(8.5–11.8)	77	5.1	(4.0–6.3)	113	5.6	(4.6–6.6)
Other ^2^	54	4.4	(3.3–5.6)	160	10.6	(9.1–12.2)	239	11.8	(10.4–13.3)
Area of residence									
Urban	889	71.6	(69.0–74.0)	1021	67.5	(65.0–69.8)	1559	77.1	(75.1–78.8)
Rural	352	28.4	(25.9–30.9)	492	32.5	(30.2–34.9)	463	22.1	(21.1–24.7)
Season of the year									
Spring/Summer	729	58.7	(54.0–61.2)	812	53.7	(50.1–57.4)	1112	55.0	(52.1–58.7)
Autumn/Winter	512	41.3	(39.5–42.5)	701	46.3	(43.8–49.7)	910	45.0	(40.1–48.7)

^1^ Sharps and fall trauma; ^2^ on a bridge, unspecified.

**Table 2 ijerph-21-00277-t002:** Standardized mortality rate for suicides according to 5-year time intervals in Paraguay between 2005 and 2019.

	Year 2005–2009			Year 2010–2014			Year 2015–2019		
Sex	Numbers	Age Rate Per 100 Thousand	Standard Age Rate Per 100 Thousand	Numbers	Age Rate Per 100 Thousand	Standard Age Rate Per 100 Thousand	Numbers	Age Rate Per 100 Thousand	Standard Age Rate Per 100 Thousand
Men	828	5.6	6.1	1031	6.5	6.7	1419	8.3	8.5
Women	347	2.4	2.4	403	2.7	2.6	487	2.9	2.9
TOTAL	1175	4.7	4.3	1434	4.6	4.7	1906	5.6	5.7

**Table 3 ijerph-21-00277-t003:** Rate of years of potential life lost by suicide according to five years by age group under 70 years in Paraguay between 2005 and 2019.

Five-Year Periods	2005–2009			2010–2014			2015–2019		
Sex	Number of Deaths	YPLL ^1^ Number	YPLL Rate ^2^	Number of Deaths	YPLL Number	YPLL Rate	Number of Deaths	YPLL Number	YPLL Rate
Men	828	30,905	2.1	1031	39,352	2.5	1419	51,562	3.02
Women	347	14,802	1.0	403	16,627	1.1	487	19,362	1.1
Total	1175	45,707	1.5	1434	55,980	1.8	1906	70,925	2.1

^1^ Years of Potential Life Lost; ^2^ per 1000.

## Data Availability

The authors do not have permission to share their study data.

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
