# Peer review of "Epidemiology of Suicide Mortality in Paraguay from 2005 to 2019: A Descriptive Study"

_ijerph, 2024, doi:10.3390/ijerph21030277_

Round 1

Reviewer 1 Report

Comments and Suggestions for Authors

I read the proposed manuscript with interest. Suicide is a major public health problem and descriptive studies investigating its characteristics are welcome. In this case, the geographical area described is Paraguay. Some comments:

- The authors often use the term "strangulation" and occasionally "hanging". What do they mean? Were there cases of suicide by strangulation other than hanging?

- By the term suffocation, do the authors mean plastic bag suffocation? 

- Line 50. The term race should be replaced with another (i.e. biogeographical origin)

- Results: Did the authors record cases of complex suicides (two different suicide modes)? This would be important data to report in the paper

- Figure 1 has non-English writing

- There are some typos that should be corrected (e.g. line 218: mal instead of men)

- The average age of suicides was very low among both males (20-24 years) and females (15-19). This is very alarming and should be discussed in detail.  

- Among the limitations of the study is also the fact that the authors did not take into consideration whether the victims were affected by medical conditions.

In general, the manuscript could be improved after a revision.

Comments on the Quality of English Language

There are some typos that should be corrected (e.g. line 218: mal instead of men)

Author Response

Comment#1: The authors often use the term "strangulation" and occasionally "hanging". What do they mean? Were there cases of suicide by strangulation other than hanging?

Response#1: The ICD-10 code - X70 specifies Intentional self-inflicted injury by hanging, strangulation or suffocation, in this study the term hanging will be used (106). We have clarified this now in the methods section. We have also revised the manuscript and decided to refer to “hanging” through the manuscript. 

Comment#2: By the term suffocation, do the authors mean plastic bag suffocation? 

Response#2: The term "asphyxiation" refers to suffocation or strangulation by means of ropes, as well as self-inflicted injuries, excluding those caused using plastic bags.

Comment#3: Line 50. The term race should be replaced with another (i.e. biogeographical origin)

Response#3: We have replaced the word “race” by “biogeographical origin” as suggested by the reviewer throughout the manuscript.

Comment#4: Results: Did the authors record cases of complex suicides (two different suicide modes)?

Response#4: We agree that information on cases of complex suicides may provide important information. Unfortunately, this information was not available. We have added the following sentence to the limitations section of this manuscript to address this:

“Finally, information on cases of complex suicides such as two different suicide modes was not available.”

Comment#5: Figure 1 has non-English writing.

Response#5: We have revised Figure 1 accordingly providing now all text in English.

Comment#6: There are some typos that should be corrected (e.g. line 218: mal instead of men).

Response#6: We have corrected the typos throughout the manuscript.

Comment#7: The average age of suicides was very low among both males (20-24 years) and females (15-19). This is very alarming and should be discussed in detail.  

Response#7: There may be a misunderstanding on the interpretation of the data in Table 1. We did not present the average age of the suicide cases. What we present in Table 1 is the frequency distribution of the age groups according to the three cohorts. The highest percentage of suicide cases were observed in the 15-24 and the 25-44 years-old population groups. This is consistent with the current literature.

Comment#8: Among the limitations of the study is also the fact that the authors did not take into consideration whether the victims were affected by medical conditions.

Response#8: We agree with the reviewer that this may be an important limitation and have addressed this now in the limitation section. The following sentence was added:

“Moreover, one limitation of this study is that we did not have information on medical conditions of the suicide cases as the data was obtained from the Vital Statistics Information Subsystem that only register the basic cause of deaths.”

Reviewer 2 Report

Comments and Suggestions for Authors

Thanks for sending the paper. It is an interesting paper. I suggest some minor changes

there are typos in the manuscript. Please have a close look and correct those.

Regarding method: Hanging is preferable to strangulation in case of suicide 

Please avoid the term victim for suicide death it creates a negative attitude. 

Author Response

Comment#1: Thanks for sending the paper. It is an interesting paper. I suggest some minor changes as there are typos in the manuscript. Please have a close look and correct those.

Response#1: Thank you. We have corrected the typos throughout the manuscript.

Comment#2: Regarding method: Hanging is preferable to strangulation in case of suicide. Please avoid the term victim for suicide death it creates a negative attitude.

Response#2: The ICD-10 code - X70 specifies Intentional self-inflicted injury by hanging, strangulation or suffocation, in this study the term hanging will be used (106). We have clarified this now in the methods section.

We have also revised the manuscript and decided to refer to “hanging” instead of strangulation. through the manuscript.

We have also removed the term “victim” from the manuscript in agreement with your suggestions.

Reviewer 3 Report

Comments and Suggestions for Authors

Thank you for the possibility of reviewing the manuscript that describes the suicide problem in Paraguay.

I think that the manuscript is fairly written.

I have small doubts which I ask the authors to consider:

In the abstract, it is mentioned that data between 2005 and 2019 was analyzed.

The authors could mention the reason for using that specific period and later in the discussion could focus on the possible impact of the cohort effect.

While providing the information about % the number of cases should be added as well.

Probably the hanging is the more proper word for describing the method of suicide. Since the mechanism can be different. Because of that, I recommend using the hanging term.

There is a lack of spacebar between from and 2005: “from2005”.

This sentence is conflicting and it has to be paraphrased:

“Decreasing trends in suicide mortality were observed in most countries across the world. Unfortunately, the mortality of suicide shows an increasing trend in many countries and populations [4].

The state that is very important for a subgroup of people who commit suicide is depression. Could authors provide information about the epidemiology of depression in Paraguay and implement it in the discussion section?

Good luck

Comments on the Quality of English Language

Minor corrections required

Author Response

Comment#1: In the abstract, it is mentioned that data between 2005 and 2019 was analyzed. The authors could mention the reason for using that specific period and later in the discussion could focus on the possible impact of the cohort effect.

Response#1: We analyzed that period since at the time of the analysis this was the time span mortality data was available, verified and entered the registry (June 2019). We added following information to the text:

Line 90: “All data between 20025 and June 2019 was analyzed as at the time of the analysis this was the time span mortality data was available, cleaned, validated and fully entered into the registry.”

We also added the following sentence in regard the cohort effect to the discussion section:

Line 224: “To some extent, we cannot rule out the cohort effect. The variation over time among the different age cohorts of the suicide cases may mask or distort effects which might be present in smaller, more homogeneous, constituent sub-cohorts as they may experience different rates of outcomes.”

Comment#2: While providing the information about % the number of cases should be added as well.

Response#2: We have added now also the number of cases to the text as recommended.

Comment#3: Probably the hanging is the more proper word for describing the method of suicide. Since the mechanism can be different. Because of that, I recommend using the hanging term.

Response#3: The ICD-10 code - X70 specifies Intentional self-inflicted injury by hanging, strangulation or suffocation, in this study the term hanging will be used (106). We have clarified this now in the methods section. We have also revised the manuscript and decided to refer to “hanging” instead of strangulation. through the manuscript.

Comment#4: There is a lack of spacebar between from and 2005: “from2005”.

Response#4: We have added a spacebar between “from” and “ 2005”.

Comment#5: This sentence is conflicting and it has to be paraphrased: “Decreasing trends in suicide mortality were observed in most countries across the world. Unfortunately, the mortality of suicide shows an increasing trend in many countries and populations [4].

Response#5: In agreement with the reviewer, we have revised the sentence. The sentence reads now as follows:

 “Even though a few countries have observed a decreasing trend in suicide mortality, data from most part of the world show an increasing trend in deaths due to suicide [4].”

Comment#6: The state that is very important for a subgroup of people who commit suicide is depression. Could authors provide information about the epidemiology of depression in Paraguay and implement it in the discussion section?

Response#6: We have added the following paragraph to the discussion to discuss the epidemiology of depression in Paraguay:

“We also would like to point out that little information exists on the prevalence of depression in the Paraguayan population. A study among a representative sample of medical students at the National University in Asuncion revealed that 19% suffered from depression [50]. Torales et al. analyzed in 2016 all psychiatric consultations carried out over one year at the Psychiatry Unit of the Emergency Department at the National University of Asunción’s Clinical Hospital in Paraguay to describe the clinical and demographic characteristics of patients and reported that approximately 17% of the patient consultations were related to depressive symptoms [51]. Finally, the Paraguayan burden of disease study found that mental behavioral disorders were the number one contributor of years of healthy life are lost with 46.4 years per 1000 inhabitants in 2019 [52]. There is a clear necessity to develop population studies to collect information on mental health disorders in the population to receive a more up-to-date information on the socio-demographic characteristics of mental health problems in Paraguay.”

Following references were added:

  1. Torales, J.; Girala, N.; Moreno, M.; Arce, A.; Trinidad, S.; Estigarribia, E.; Vera, J. J.; Mogelós, D.; Cáceres, M.; Velázquez, F.; Benítez, D.; Ocampos, S.; Saua, A. Depresión y ansiedad en estudiantes de medicina de la Universidad Nacional de Asunción. Revista Paraguaya de Psiquiatría 2013, 1(1), 7-23.
  2. Torales, J.; Ventriglio, A.; Barrios, I.; Arce, A. Demographic and clinical characteristics of patients referred to the psychiatry unit of the emergency department at the National University of Asunción’s General Hospital, Paraguay. International Journal of Culture and Mental Health 2016, 9(3), 233-238, doiI: 10.1080/17542863.2016.1197290
  3. Ministerio de Salud Publica y Bienestar Social. Carga de enfermedad en Paraguay. Estimación de los años de vida saludables perdidos 2019. Available at: https://dgvs.mspbs.gov.py/wp-content/uploads/2023/08/Informe-CARGA-de-la-enfermedad-publicacion-DGVS.pdf [accessed 19/2/2024]